# Omega-3 Long-Chain Polyunsaturated Fatty Acids Intake by Ethnicity, Income, and Education Level in the United States: NHANES 2003–2014

**DOI:** 10.3390/nu12072045

**Published:** 2020-07-09

**Authors:** Caleb Cave, Nicholas Hein, Lynette M. Smith, Ann Anderson-Berry, Chesney K. Richter, Karl Stessy Bisselou, Adams Kusi Appiah, Penny Kris-Etherton, Ann C. Skulas-Ray, Maranda Thompson, Tara M. Nordgren, Corrine Hanson, Melissa Thoene

**Affiliations:** 1Department of Pediatrics, University of Nebraska Medical Center, Omaha, NE 68198, USA; caleb.cave@unmc.edu (C.C.); alanders@unmc.edu (A.A.-B.); maranda.thompson@unmc.edu (M.T.); 2College of Public Health, University of Nebraska Medical Center, Omaha, NE 68198, USA; nickahein@gmail.com (N.H.); lmsmith@unmc.edu (L.M.S.); karlstessy.bisselou@unmc.edu (K.S.B.); adams.kusiappiah@unmc.edu (A.K.A.); 3Department of Nutritional Sciences, University of Arizona, Tucson, AZ 85721, USA; richterck@email.arizona.edu (C.K.R.); skulasray@email.arizona.edu (A.C.S.-R.); 4Department of Nutritional Sciences, The Pennsylvania State University, University Park, PA 16802, USA; pmk3@psu.edu; 5Division of Biomedical Sciences, School of Medicine, University of California Riverside, Riverside, CA 92521, USA; 6College of Allied Health Professions, University of Nebraska Medical Center, Omaha, NE 68198, USA; ckhanson@unmc.edu

**Keywords:** omega-3 fatty acids, eicosapentaenoic acid, docosahexaenoic acid, oily fish, fish oil supplements, ethnicity, income, education

## Abstract

Although there are many recognized health benefits for the consumption of omega-3 (n-3) long-chain polyunsaturated fatty acids (LCPUFA), intake in the United States remains below recommended amounts. This analysis was designed to provide an updated assessment of fish and n-3 LCPUFA intake (eicosapentaenoic (EPA), docosahexaenoic acid (DHA), and EPA+DHA) in the United States adult population, based on education, income, and race/ethnicity, using data from the 2003–2014 National Health and Nutrition Examination Survey (NHANES) (*n* = 44,585). Over this survey period, participants with less education and lower income had significantly lower n-3 LCPUFA intakes and fish intakes (*p* < 0.001 for all between group comparisons). N-3 LCPUFA intake differed significantly according to ethnicity (*p* < 0.001), with the highest intake of n-3 LCPUFA and fish in individuals in the “Other” category (including Asian Americans). Supplement use increased EPA + DHA intake, but only 7.4% of individuals consistently took supplements. Overall, n-3 LCPUFA intake in this study population was low, but our findings indicate that individuals with lower educational attainment and income are at even higher risk of lower n-3 LCPUFA and fish intake.

## 1. Introduction

Omega-3 (n-3) long-chain polyunsaturated fatty acids (LCPUFA), particularly eicosapentaenoic (EPA) and docosahexaenoic acid (DHA), can offer many health benefits when regularly consumed in sufficient quantities. Health benefits of n-3 LCPUFA include reduction of preterm birth, decreased risk for low birth weight in infants, improved visual acuity in infants, facilitation of early childhood neurodevelopment, inflammation modulation such as with chronic disease or cancer-related complications, risk reduction of cardiovascular disease (CVD), prevention of dementia and cognitive decline, and a decreased risk for developing age-related macular degeneration [1,2,3,4,5,6,7,8,9,10,11,12,13].

There are currently no Recommended Dietary Allowances or Dietary Reference Intakes set for EPA or DHA. Other government and public health agencies have offered recommendations, but these vary widely and are based primarily on age or gender [14,15,16]. Although symptomatic polyunsaturated fatty acid deficiency, which presents with scaly skin and dermatitis, is very rare in the United States (US), the Western diet typically includes lower amounts of n-3 LCPUFA than those recommended for age and gender groups by the National Institutes of Health and the World Health Organization [14,17,18,19,20] (Table 1).

EPA and DHA are the most common forms of n-3 LCPUFA in the human diet and are primarily found in fatty, cold-water fish and other seafood. Alpha-linolenic acid (ALA), the precursor to EPA and DHA, is an essential n-3 fatty acid that cannot be synthesized in the human body [21] and must be obtained from the diet—primarily from plant-based sources such as flaxseed oil, canola oil, chia seeds, and walnuts [22]. Conversion of ALA to EPA and DHA in adults is extremely low (<10%), and may also differ by sex [23,24]. Therefore, fatty fish is considered the primary source of n-3 LCPUFA in the human diet [22]. Supplementation with fish oil, krill oil, or products derived from algal sources can also provide substantial amounts of n-3 LCPUFA [20].

Differences in n-3 LCPUFA intake exist among subgroups of the US population. For example, our recent analysis of NHANES (National Health and Nutrition Examination Survey) 2003–2014 data revealed differences in n-3 LCPUFA intake based on age, gender, and pregnancy status [25]. Overall, n-3 LCPUFA intake was significantly lower in women compared to men, younger participants (regardless of gender), and pregnant women (compared to their non-pregnant counterparts) [25]. Other analyses of NHANES data from 2003 to 2014 have further demonstrated lower n-3 LCPUFA intakes in pregnant women or women of child-bearing age and younger participants (compared to their male and older counterparts) [26,27]. These recent findings strongly suggest that n-3 LCPUFA intake is influenced by factors such as gender and pregnancy status.

Much less is known about the potential role of other demographic factors on n-3 LCPUFA intake, such as ethnicity, income, and education level. A recent analysis of NHANES 2011–2014 data suggested that n-3 LCPUFA intake varies by ethnicity in the US, with non-Hispanic Asians consuming significantly more EPA and DHA than Hispanics, non-Hispanic Whites, and non-Hispanic Blacks [28]. However, in NHANES 2005–2010, overall fish intake in the US did not differ based on race/ethnicity but was directly correlated with income and education [29]. To our knowledge, this relationship has not yet been described for intake of fish high in n-3 LCPUFA. Additionally, no similar analyses of NHANES data spanning more than a 5-year period have been conducted and additional research is needed to further evaluate the relationship between n-3 LCPUFA intake with ethnicity, income, and education.

The purpose of this study was to identify subgroups of the US population that are at greater risk for low n-3 LCPUFA intake (EPA, DHA, and EPA + DHA) by comparing differences in current intake according to education level, ethnicity, and income. Fish intake (both overall fish intake and intake of fish high in n-3 LCPUFA, EPA and DHA) and supplement use were also compared among these population subgroups.

## 2. Materials and Methods

Demographics and dietary interviews from six cycles of NHANES (2003–2004, 2005–2006, 2007–2008, 2009–2010, 2011–2012, and 2013–2014) were used for the analysis. Data on total nutrient intake were collected from two 24 h dietary recall interviews. Dietary supplement use was assessed over a 30-day period. EPA and DHA intake from supplements was calculated using an external list of over-the-counter supplements, as described by Thompson et al. [25]. All statistical analyses were performed in R (version 3.4.2; R Core Team, 2017; Vienna, Austria) using the SURVEY package [30,31,32]. Statistical significance was set at *p* < 0.05.

NHANES uses a complex survey design and represents the civilian, noninstitutionalized US population. Detailed methodology of NHANES has been described elsewhere [33]. Dietary sampling weights were used for all analyses and adjusted to account for multiple cycles. All individuals aged 1 year or older were included in the analysis, except for pregnant females. Additionally, adults (19 years or older) were excluded if their daily caloric intake was <800 or >8000 for males and <600 or >6000 for females. Caloric intake was calculated as the average caloric intake from the 1st and 2nd day of 24-h dietary recalls. If the 2nd day was missing, the average caloric intake was set to the 1st day.

Demographics of the sample are reported as frequencies for categorical variables and as means with standard error of the mean for continuous variables. Race was categorized as non-Hispanic White, non-Hispanic Black, Mexican-American, Other Hispanic, and Other. Education was categorized as <high school and <19 years old, <high school and ≥19 years old, completed high school, and completed school beyond high school. Poverty-Income Ratio (PIR) was dichotomized at the median of the sample (i.e., 2.73) and also at the federal poverty line (FPL) (i.e., 1). EPA and DHA intakes were calculated as the average of the 1st and 2nd day values from the 24-h dietary recall. As with caloric intake, if the 2nd day was missing, only the 1st day was used. EPA and DHA were further adjusted for average caloric intake. Total fish intake in the last 30 days was calculated based on the number of times a respondent reported eating each of the 31 types of fish included in the NHANES survey. The number of times a particular fish was consumed in the last 30 days was assumed to be zero if an individual responded “No” for eating that fish. Intake of fish high in n-3 LCPUFA was calculated using the above-described method, but limited to 7 fish only (tuna, mackerel, salmon, sardines, shark, swordfish, and trout), similar to previous NHANES analysis of n-3 LCPUFA intake [25]. Comparisons of EPA/DHA (milligrams/1000 kilocalorie or mg/1000 kcal) intake and fish eaten in the last 30 days by race, education, and income were made by fitting a generalized linear model and using a Rao–Scott test to determine whether all coefficients were zero. N-3 LCPUFA intakes (mg) were adjusted per 1000 kcal to control for variation in individual energy requirements based on sex, physical activity, age, and/or body size.

EPA/DHA intake (mg/1000 kcal) in supplement users and non-supplement users was compared using a t-test. Supplement users were further dichotomized as taking a supplement with or without EPA/DHA. For individuals taking a supplement containing EPA/DHA, total EPA/DHA intake was calculated using the label information provided for each over-the-counter supplement, as previously described by Thompson et al. [25]. The mean intake of EPA/DHA (mg/1000 kcal) was then compared among non-supplement users, individuals taking a supplement without EPA/DHA, and individuals taking a supplement with EPA/DHA by fitting a generalized linear model and using a Rao–Scott test to determine whether all coefficients were zero.

## 3. Results

### 3.1. Participant Characteristics

Characteristics of the participant population (*n* = 44,585) and average EPA, DHA, and EPA+DHA intakes are provided in Table 2, according to specified population subgroups.

### 3.2. EPA, DHA, and EPA+DHA Intake by Ethnicity, Education Level, and Income

EPA, DHA, and EPA+DHA intake among all ethnic groups included in the analysis are shown in Figure 1. EPA and EPA+DHA intake among ethnicities followed a specific trend: Other > Non-Hispanic Black > Other Hispanic > Non-Hispanic White > Mexican American.

Regarding educational attainment, participants with greater than high school education had significantly higher dietary intake of EPA and combined EPA+DHA than those with a high school education or lower than a high school education (*p* < 0.05; Figure 2). Participants who graduated from high school and those who did not finish high school and were ≥19 years old consumed significantly more EPA, DHA, and EPA+DHA than participants who did not finish high school and were <19 years old (*p* < 0.0001).

Daily EPA and DHA intake also differed significantly according to income, whether income was stratified according to median PIR or the Federal Poverty Line (*p* < 0.001; Figure 3).

### 3.3. Fish Intake by Ethnicity, Education Level, and Income

Significant differences in the intake of total fish and fish high in n-3 LCPUFA were found across all ethnic groups (*p* < 0.001). Individuals categorized in the “Other” category had the highest mean intake of both total fish and fish high in n-3 LCPUFA (9.7 servings/month and 2.7 servings/month, respectively) while Mexican Americans had the lowest intake (5.7 servings/month and 1.6 servings/month, respectively), as shown in Figure 4. Intake of total fish as was similar between Non-Hispanic Whites, Non-Hispanic Blacks (7.6 servings/month each), and Other Hispanics (7.2 servings/month). However, Non-Hispanic Blacks had lower intake of fish high in n-3 LCPUFA (1.8 servings/month) compared to but Non-Hispanic Whites (2.6 servings/month) and Other Hispanics (2.2 servings/month).

A higher level of education was also associated with a greater number of servings of total fish and fish high in n-3 LCPUFA consumed over the course of 30 days: >high school (8.9 servings/month and 3 servings/month, respectively), high school (6.9 servings/month and 2.0 servings/month), <high school and ≥19 years old (6.3 servings/month and 1.6 servings/month), and <high school and <19 years old (5.2 servings/month and 1.5 servings/month) (Figure 5).

As with EPA and DHA intake by income level, intake of both total fish and fish high in n-3 LCPUFA was higher in participants with higher incomes, whether this was defined based on median Federal Poverty Line or median PIR (Figure 6) (all *p* < 0.001).

### 3.4. EPA/DHA-Containing Supplement Use by Ethnicity, Education Level, and Income

EPA/DHA-containing supplements were used by 7.4% of all participants. EPA and DHA intake were both significantly higher in participants who reported the use of any supplement (whether it contained n-3 LCPUFA or not) compared to those who did not report any supplement use (Figure 7 and Figure 8; all *p* < 0.001). Participants taking supplements containing EPA/DHA had significantly higher intake of EPA and DHA per 1000 kcal (163.9 mg and 144.5 mg, respectively) when compared to those taking supplements without EPA/DHA (16.6 mg and 33.1 mg, respectively) and those not taking any supplements at all (14.9 mg and 29.7 mg, respectively) (all *p* < 0.001). EPA/DHA supplement use increased with both education and income level (Table 3), matching the trend found for EPA+DHA and fish intake. As for EPA/DHA supplement use by ethnicity, supplement use was highest in Non-Hispanic Whites (6.0%) and below 0.5% of the population for other subgroups.

## 4. Discussion

Our analysis of NHANES 2003–2014 demonstrates that there are significant differences in n-3 LCPUFA intake among the US population based on ethnicity, education level, and income. Overall, EPA/DHA intake, as well as supplement use and the consumption of fish high in n-3 LCPUFA, increased in conjunction with education and income. This consistent pattern suggests that socioeconomically disadvantaged populations within the US are at heightened risk for lower n-3 LCPUFA intake, which may further contribute to health disparities.

Previous analyses have indicated similar disparities in n-3 LCPUFA intake. In an analysis of NHANES 2011–2014 data, overall supplement use by adults was higher in participants who were well above the poverty level, were food-secure, and did not participate in SNAP (Supplemental Nutritional Assistance Program)—regardless of age, gender, and ethnicity [34]. A similar trend has also been illustrated in Australia, where an analysis of the national Australian Health Survey revealed that unemployed citizens with less educational attainment tended to consume less fish overall, with a higher proportion of fish that are lower in n-3 LCPUFA content, when compared to higher-educated, higher-earning counterparts [35]. This pattern may be attributed to many factors, including lack of access to information concerning proper nutrition, as well as an overall lack of resources. This could have important implications for future generations as adequate n-3 LCPUFA intake is essential for promoting early neurodevelopment [6,7,36,37]. Studies have also suggested improved memory and attention among children and adolescents with higher prenatal or current intake of n-3 LCPUFA [38,39,40]. A causal relationship between n-3 LCPUFA intake and education or income cannot be derived from our analysis, but these results raise the question of how socioeconomic disadvantage influences early-life n-3 LCPUFA intake and if this leads to additional health and developmental consequences with long-term implications.

Globally, the highest blood concentrations of n-3 LCPUFA have been found in Japan and other Asian countries, while the lowest levels are in North America, Central and South America, and most African and European countries [41]. Similarly, differences in dietary intake of n-3 LCPUFA varied between ethnic groups in our analysis. For example, EPA/DHA intake and intake of fish high in n-3 LCPUFA were consistently highest in participants categorized as “Other” for ethnicity, which includes Asian-Americans. This is consistent with analyses of NHANES 2011–2014 that found non-Hispanic Asians consumed 2× more DHA and 3 × more EPA than other ethnic groups, with oily fish rich in n-3 LCPUFA being the highest contributor to EPA and DHA intake [28]. Interestingly, non-Hispanic Blacks had the second highest intake of EPA and DHA in our analysis, but the second lowest intake of fish high in n-3 LCPUFA, indicating that this subpopulation’s primary dietary sources of n-3 LCPUFA are not the standard sources highest in EPA and DHA (i.e., fatty, cold-water fish such as salmon, mackerel, tuna, and sardines). Further research is needed to identify the primary sources of n-3 LCPUFA intake in this population, but it is possible that this is due to the consumption of foods fortified with n-3 LCPUFA.

With regard to supplement use and n-3 LCPUFA intake, reported use of EPA/DHA-containing supplements was low in all ethnicity categories. Notably, despite having the highest use of EPA/DHA supplements, non-Hispanic Whites still had the 2nd lowest overall n-3 LCPUFA intake of any ethnic group. Thus, although EPA/DHA supplements are a potentially viable alternative to dietary sources of n-3 LCPUFA, the small number of individuals using these supplements does not significantly increase the overall n-3 LCPUFA intake of the subpopulation. In addition to supplements, fortified foods, and other alternatives may be beneficial and deserve further consideration as a means of increasing n-3 LCPUFA intake in the US.

Compared to EPA + DHA intakes recommended by the World Health Organization (200–250 mg/day for adults) and the National Institutes of Health (270 mg/day), average intake in this US population remained far below recommended amounts at an average daily mean of approximately 100 mg [14,18]. Oily fish are the primary dietary source of n-3 LCPUFA, but very few demographic groups in our analysis met the American Heart Association’s suggested intake of 2 servings/week of fish for adults or 8 ounces/week of oily fish per the 2010 Dietary Guidelines for Americans (provides approximately 250 mg/day EPA + DHA) [42]. Additionally, although n-3 LCPUFA supplements are commonly available and can significantly augment overall EPA + DHA intake, our findings and previous research shows that only a small proportion of the population routinely uses these supplements [27]. Education and intervention approaches to increase n-3 LCPUFA intake should include a variety of strategies, including increased consumption of dietary n-3 LCPUFA sources, consistent daily use for those who choose to use EPA/DHA supplements, as well as n-3 LCPUFA fortified foods and other potential alternatives.

### Strengths and Limitations

This study analyzed a large, representative sample of the US population, using a 24 h dietary recall method that is considered sufficient for accurately measuring mean dietary intake at the population level [43]. We stratified our analysis by ethnicity, education, and income but did not consider differences by gender or age. We did not assess other factors that may influence n-3 LCPUFA intake (e.g., age, gender, pregnancy, geographic regions), but prior studies have already revealed significant differences based on several of these population traits [25,26,44]. Lastly, we did not stratify education or income level by ethnicity, which could also reveal significant disparities in n-3 LCPUFA intake. Much less is known about docosapentaenoic acid (DPA), the metabolic intermediate of EPA and DHA, but existing evidence indicates that DPA may also contribute to the health benefits attributed to EPA and DHA [45,46,47]. Although we did not assess differences in dietary intake DPA alongside EPA and DHA in this analysis, we previously reported differences in DPA intake according to age, gender, and race/ethnicity in this NHANES 2003–2014 cohort [27]. It should also be noted that 24 h dietary recalls may not accurately reflect an individual’s habitual dietary pattern, and a recent study identified discrepancies in the NHANES dataset regarding energy intake values [48]. To reduce the potential effect of this, we limited our study to participants who reported plausible caloric intake ranges and excluded adult men with caloric intake of <800 kcal or >8000 kcal per day and adult women with caloric intake of <600 kcal or >6000 kcal per day. Furthermore, we also included an assessment of fish intake over a 30-day period that corroborated and expanded upon our overall results for EPA and DHA intake from 24 h dietary recalls. However, assessing n-3 LCPUFA intake based on fish consumption also has limitations as the EPA and DHA content of a particular species of fish can differ based on its source, with some species of farmed fish having substantially lower EPA and DHA content than their wild-caught counterparts [49]. Nonetheless, our results agree with other recent investigations, providing consistent evidence for interpretation [20,28,41]. The cross-sectional nature of NHANES must also be considered in the interpretation of our results, as we cannot establish any causal relationships related to intake. Additionally, there remains a lack of standardized guidelines for sufficient n-3 LCPUFA intake and research regarding the amount of n-3 LCPUFA required to support specific physiological functions remains limited. This makes the development of evidence-based recommendations for EPA and DHA intake difficult.

## 5. Conclusions

In this study, we assessed differences in EPA and DHA intake in the US according to ethnicity, education, and income using 2003–2014 NHANES data. Overall, EPA+DHA intake was low, with an average daily intake of 100 mg. Low educational attainment and income were consistently associated with lower n-3 LCPUFA intake. With regard to ethnicity, average n-3 LCPUFA intake was highest in the “Other” category (including Asian-Americans). Fish intake was also below recommended amounts in all demographic subgroups, except for individuals categorized as “Other” and individuals with a higher level of education and income. Use of EPA/DHA supplements was low in all population subgroups. Consumption of n-3 LCPUFA remains low in the overall US population, with even lower n-3 LCPUFA intake being more common in socioeconomically disadvantaged subpopulations and specific ethnicities, which may indicate the potential for augmenting health disparities.

## Figures and Tables

**Figure 1 nutrients-12-02045-f001:**
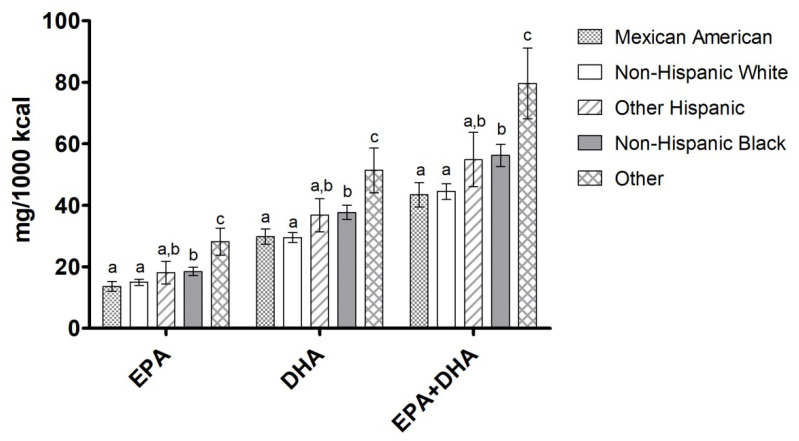
Omega-3 LCPUFA intake (mg per 1000 kcal) by race/ethnicity. Data are presented as unadjusted means ± SE and were compared using the SURVEY package in R (version 3.4.2). Categories annotated with different letters indicate a significant difference between race/ethnicity groups in post hoc pairwise comparisons (*p* < 0.05).

**Figure 2 nutrients-12-02045-f002:**
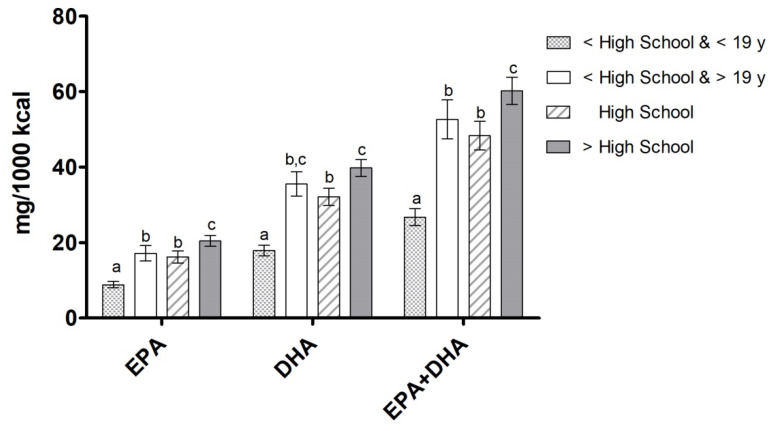
Omega-3 LCPUFA intake (mg per 1000 kcal) by education level. Data are presented as unadjusted means ± SE and were compared using the SURVEY package in R (version 3.4.2). Categories annotated with different letters indicate a significant difference between education groups in post hoc pairwise comparisons (*p* < 0.05).

**Figure 3 nutrients-12-02045-f003:**
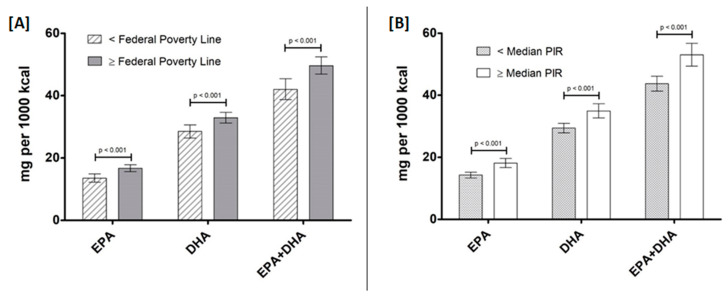
Omega-3 LCPUFA intake (mg per 1000 kcal) by Federal Poverty Line (**A**) and Poverty Income Ratio (PIR) (**B**). Data are presented as unadjusted means ± SE and were compared using the SURVEY package in R (version 3.4.2). *p*-values represent within group comparisons for each n-3 LCPUFA.

**Figure 4 nutrients-12-02045-f004:**
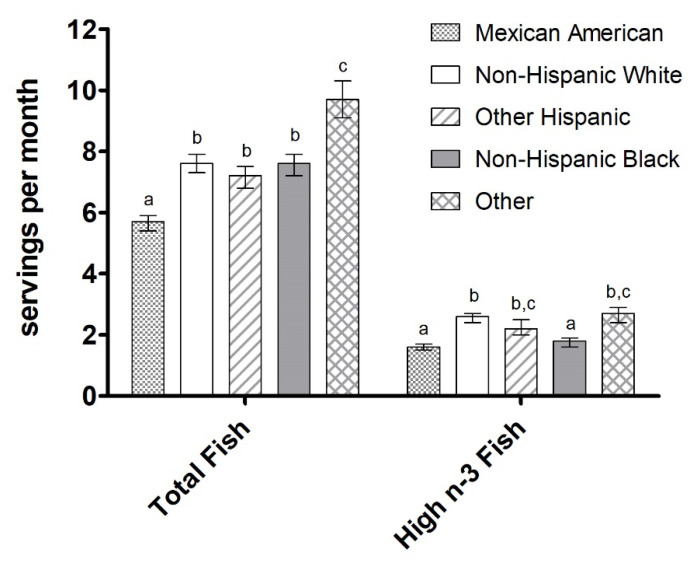
Total fish intake and intake of fish high in n-3 LCPUFA (servings/month) by race/ethnicity. Data are presented as unadjusted means ± SE and were compared using the SURVEY package in R (version 3.4.2). Categories annotated with different letters indicate a significant difference between race/ethnicity groups in post hoc pairwise comparisons (*p* < 0.05).

**Figure 5 nutrients-12-02045-f005:**
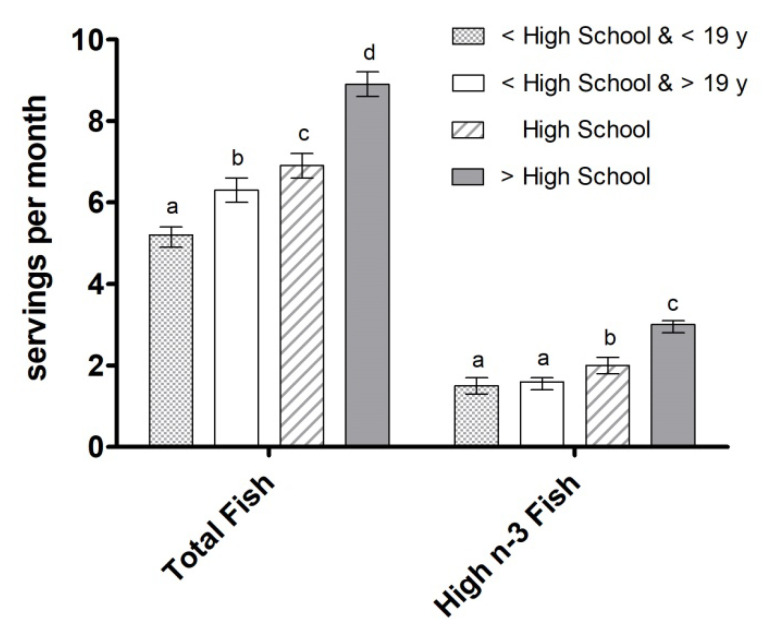
Total fish intake and intake of fish high in n-3 LCPUFA (servings/month) by education level. Data are presented as unadjusted means ± SE and were compared using the SURVEY package in R (version 3.4.2). Categories annotated with different letters indicate a significant difference between education groups in post hoc pairwise comparisons (*p* < 0.05).

**Figure 6 nutrients-12-02045-f006:**
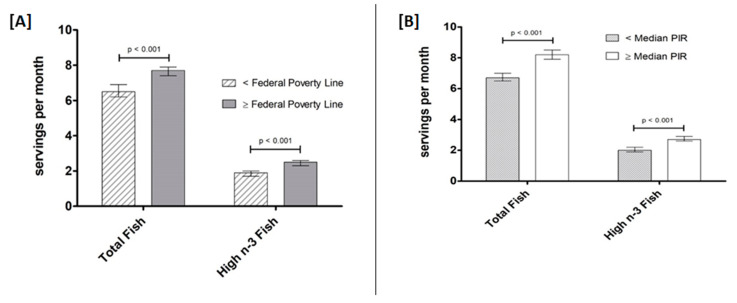
Total fish intake and intake of fish high in n-3 LCPUFA (servings/month) by Federal Poverty Line (**A**) and Poverty Income Ratio (PIR) (**B**). Data are presented as unadjusted means ± SE and were compared using the SURVEY package in R (version 3.4.2). P-values represent within-group comparisons for total fish and fish high in n-3 LCPUFA.

**Figure 7 nutrients-12-02045-f007:**
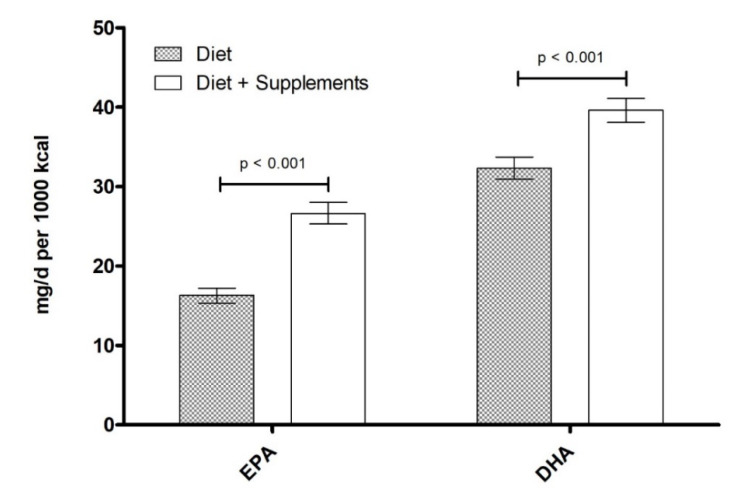
EPA/DHA average daily intake (mg per 1000 kcal) categorized by diet only and diet + supplement use. Data are presented as unadjusted means ± SE and were compared using the SURVEY package in R (version 3.4.2). *p*-values represent within group comparisons for EPA and DHA.

**Figure 8 nutrients-12-02045-f008:**
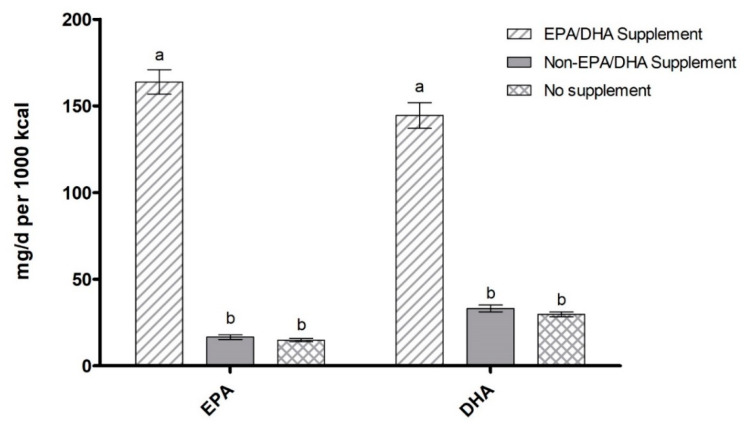
EPA/DHA average daily intake (mg per 1000 kcal) categorized by supplement use. Data are presented as unadjusted means ± SE and were compared using the SURVEY package in R (version 3.4.2). Categories annotated with different letters indicate significant within-group differences in post hoc pairwise comparisons (*p* < 0.05).

**Table 1 nutrients-12-02045-t001:** Adequate Intakes for omega-3 (n-3) long-chain polyunsaturated fatty acids (LCPUFA).

**World Health Organization** [14]	**Age**	**AI (Adequate Intake; per day)**
DHA	12–24 months	10–12 mg/kg
EPA+DHA	2–4 years	100–150 mg
	4–6 years	150–200 mg
	6–10 years	200–250 mg
	Adults	200–250 mg
**National Institutes of Health** [18]	**Age**	**Male/Female (per day)**
EPA+DHA	1–3 years	70 mg/70 mg
	4–8 years	90 mg/90 mg
	9–13 years	120 mg/100 mg
	14–18 years	160 mg/110 mg
	19–50 years	160 mg/110 mg
	51 years	160 mg/110 mg

mg = milligrams, kg = kilograms.

**Table 2 nutrients-12-02045-t002:** Descriptive statistics of the 2003–2014 National Health and Nutrition Examination Survey (NHANES) study population (*n* = 44,585).

**Characteristics**		**N**	**Mean (SE)**
Age (years)		44,585	37.3 (22.1)
Calories (kcal)		44,482	2064 (793.6)
Poverty-Income Ratio		41,692	2.9 (1.7)
EPA intake (mg/1000 kcal)		44,482	16.3 (44.4)
DHA intake (mg/1000 kcal)		44,482	32.3 (68.6)
EPA+DHA intake (mg/1000 kcal)		44,482	48.5 (110.9)
Total fish intake (servings/30 days)		29,033	7.5 (6.4)
Fish high in n-3 LCPUFA intake (servings/30 days)		24,909	2.4 (3.4)
Fish low in n-3 LCPUFA intake (servings/30 days)		29,033	5.4 (4.7)
**Characteristics**		**N**	**%**
Gender	Male	22,056	49.0
	Female	22,529	51.0
Race/Ethnicity	Non-Hispanic Black	10,453	12.0
	Non-Hispanic White	17,809	66.7
	Mexican-American	9212	9.9
	Other Hispanic	3588	5.0
	Other	3523	6.4
Education	<high school and <19 years old	12,442	19.3
	<high school	6370	13.0
	high school	6601	19.5
	>high school	13,645	48.2
Income	<median PIR	26,602	49.8
	≥median PIR	15,090	50.2
	<FPL	10,770	16.6
	≥FPL	30,922	83.4

(PIR = Poverty-Income Ratio. Median PIR =2.73. FPL = Federal Poverty Line).

**Table 3 nutrients-12-02045-t003:** Proportion of participants taking a supplement containing EPA/DHA and probability of use by education, race/ethnicity, and income.

		Reported EPA/DHA Supplement Use (%)
Total Population		2420
Education	<high school and <19 y	148 (1.5%)
	<high school	310 (5.5%)
	High school	458 (7.3%)
	>high school	1445 (11.3%)
Race/Ethnicity	Non-Hispanic Black	330 (3.2%)
	Non-Hispanic White	1388 (9.0%)
	Mexican American	223 (3.0%)
	Other Hispanic	207 (5.4%)
	Other	272 (7.4%)
Income	<median PIR	990 (4.7%)
	≥median PIR	1262 (10.0%)
	<FPL	224 (2.6%)
	≥FPL	2028 (8.3%)

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
