# Peer review of "Omega-3 Long-Chain Polyunsaturated Fatty Acids Intake by Ethnicity, Income, and Education Level in the United States: NHANES 2003–2014"

_nutrients, 2020, doi:10.3390/nu12072045_

Round 1

Reviewer 1 Report

The authors aim was “to identify subgroups of the US population that are at greater risk for low n-3 fatty acid intake by comparing differences in current intake according to education level, ethnicity, and income. Fish intake (both overall fish intake and intake of fish high in n-3 fatty acids) and supplement use were also compared among these population subgroups.”

This is a well written manuscript although the authors need to address two concerns and fix a few errors.

Firstly, the terminology of n-3 fatty acids should align with other publications in this field. When referring to n-3 fatty acids, this definition could include alpha-linolenic acid. Given that the focus of this manuscript is on EPA and DHA intakes and they are the omega-3 long chain polyunsaturated fatty acids (n-3 LCPUFA), the manuscript should use the n-3 LCPUFA terminology throughout the manuscript, instead of n-3 fatty acids. Therefore the authors should initially define the n-3 fatty acids (i.e. EPA and DHA) as omega-3 long chain polyunsaturated fatty acids (n-3 LCPUFA) and replace all subsequent n-3 fatty acids with the abbreviated term n-3 LCPUFA throughout the manuscript. Or alternatively use EPA and DHA instead of n-3 fatty acids throughout the manuscript.

Secondly, it is unclear why the authors report the dietary intakes of EPA and DHA per energy. Why not report actual intakes without correcting for energy? The global recommendations for EPA and DHA are in mg per day without correcting for energy; thereby making comparisons easier to recommended intakes and also other publications in this area.

Moreover, oily fish contains more energy than lean fish and then to report the n-3 LCPUFA intakes per energy, would be different depending on whether oily or lean fish was consumed. As oily fish contains more energy than lean fish and then to correct n-3 LCPUFA intakes for energy will reduce the n-3 LCPUFA intakes more so if oily fish was consumed than if lean fish was consumed. This is confounding the actual n-3 LCPUFA intake per day.

For example, consumption of 100g Atlantic salmon has 2450mg n-3 LCPUFA versus 100g of Bassa fillet has 70mg n-3 LCPUFA; therefore Atlantic salmon n-3 LCPUFA is 35 times higher than Bassa fillet. The energy for 100g of Atlantic Salmon is 206 kcal therefore n-3 LCPUFA per kcal is 11.9. The energy for 100g of Bassa fillet is 68 kcal and therefore the n-3 LCPUFA per kcal is 1.09. Now the Atlantic Salmon per kcal is 11 times higher than Bassa fillet. So when the authors show the results of n-3 LCPUFA per kcal, it is highly influenced by whether or not oily fish versus lean fish was consumed. Therefore the authors should report the actual n-3 LCPUFA intakes without correcting for energy. This would mean comparison to global recommendations easier, as well as other publications that report on actual n-3 LCPUFA as mg per day without correcting for energy. Or the authors report both, i.e. actual n-3 LCPUFA in takes as mg per day and as mg per energy per day.

By reporting the actual n-3 LCPUFA in takes as mg per day would make more sense to the reader when the author state “average intake in this US population remained far below recommended amounts at an average daily mean of approximately 100 mg” lines 252-254 and “Overall, EPA+DHA intake was low, with an average daily intake of 100 mg” lines 291-292”.

The authors need to correct the following:

Lines 160-164

“Non-Hispanic Whites and Non-Hispanic Blacks (7.6 servings/month each) both had significantly higher mean intakes of total fish than Other Hispanics (7.2 servings/month), but Non-Hispanic Whites (2.6 servings/month) had significantly higher intake of fish high in n-3 fatty acids than both Other Hispanics (2.2 servings/month) and Non-Hispanic Blacks (1.8 servings/month), all p<0.001.” Some of these statements are incorrect.

“Non-Hispanic Whites and Non-Hispanic Blacks (7.6 servings/month each) both had significantly higher mean intakes of total fish than Other Hispanics (7.2 servings/month)”, - this is incorrect, there is no significant difference between non-Hispanic white, non-Hispanic black and other Hispanic (all letter ‘b’).

“but Non-Hispanic Whites (2.6 servings/month) had significantly higher intake of fish high in n-3 fatty acids than both Other Hispanics (2.2 servings/month)” - this is incorrect, there is no significant difference between non-Hispanic white and other Hispanic (they both have the letter ‘b’).

Table 1 Table 1 Recommended Dietary Intake for n-3 fatty acids is incorrect. Adequate Intake (AI): Intake at this level is assumed to ensure nutritional adequacy; established when evidence is insufficient to develop an RDA. Recommended Dietary Allowance (RDA): Average daily level of intake sufficient to meet the nutrient requirements of nearly all (97%–98%) healthy individuals; often used to plan nutritionally adequate diets for individuals. Therefore AI is not a recommended dietary intake (or RDA). The Table 1 title should be Adequate Intakes for n-3 LCPUFA.

Reviewer 2 Report

In this manuscript, the authors described ‘Omega-3 Fatty Acid Intake by Ethnicity, Income, and Education Level in the United States: NHANES 2003-2014’. Although the results are partially original, there are many weakness and flaws that require authors' further attention:

Major and Minor points:

  1. Although the authors described ‘Omega-3 Fatty Acid Intake by Ethnicity, Income, and Education Level in the United States: NHANES 2003-2014’, I think this topic is not significant for readers.
  2. The authors described to omega-3 fatty acid in the title. And also described to n-3 fatty acids in text. Although omega-3 fatty acids and n-3 fatty acids are the same material, you have to select one of them to avoid readers’ confusion. In the case of omega-3 fatty acid is ω3-PUFA and n-3 fatty acid is n-3 PUFA, respectively.
  3. As you know, omega-3 fatty acids composed of several materials. Therefore, you have to use the term as the plural. For example, omega-3 polyunsaturated acids (ω3-PUFA) or n-3 polyunsaturated fatty acids (n-3 PUFA)…..
  4. In the Introduction section, authors have to add cancer prevention and anti-inflammation as health benefits.
  5. All legends are too simple! Therefore, it is not easy to understand the results. You have to describe all legends again. And also, you have to unifyω3-PUFA or n-3 PUFA to avoid readers’ confusion in Figure and Table legends.
  6. Several Figures’ descriptions have errors! For example, on the y-axis of 1 and 2, the authors have to change mg/kcal to mg/1,000 kcal. Please check all Tables and Figures.
  7. I think that Fig. 8 is not significant. If you agree, remove it for readers.

Reviewer 3 Report

Thank you for this inteersting study. I wonder if n-3 DPA could be included, guess it's difficult as the earlier studies you are using probabaly don't include analyses of DPA. As it is an important part od the Cascade in the n-3 derivatives, could be Worth including.

Could possibly  be mentioned in you r discussion as an interesting add in future studies.

Round 2

Reviewer 2 Report

In this manuscript, the authors described on ‘Omega-3 Fatty Acid Intake by Ethnicity, Income, and Education Level in the United States: NHANES 2003-2014’. Although the revised manuscript is partially improved, there are several weaknesses and flaws that require the authors' further attention.

  1. The authors changed to omega-3 long chain polyunsaturated fatty acid. In this case, you remove the long chain. w3-PUFA is better than w3-LCPUFA!
  2. In the title, the authors did not change “Omega-3 Long-Chain Polyunsaturated Fatty Acid” to Omega-3 Polyunsaturated Fatty Acids. You have to use the word as the plural.
  3. In the Introduction section, although authors added cancer prevention and anti-inflammation as health benefits, you did not add a related-references. Therefore, you have to add them in References.
  4. Although you changed mg/kcal to mg/1,000 kcal, it has still remained in the manuscript. Please read the manuscript again and change them.
  5. In 4. EPA/DHA-Containing Supplement ----- Sections, please divide Fig. 7 and Fig. 8 and explain, respectively. I think readers will not understand them. In the y-axis of Fig. 7 and 8, you expressed mg/d per 1000 kcal. Why? How is mg/1000 kcal?
  6. All legends are still simple. Readers just read the Figure legend and have to understand it.
